# Autoimmune Rheumatic Diseases and Vascular Function: The Concept of Autoimmune Atherosclerosis

**DOI:** 10.3390/jcm10194427

**Published:** 2021-09-27

**Authors:** Ahmed M. Hedar, Martin H. Stradner, Andreas Roessler, Nandu Goswami

**Affiliations:** 1Physiology Division, Otto Loewi Center of Research in Vascular Biology, Immunity and Inflammation, Medical University of Graz, Neue Stiftingtalstraße 6, 8010 Graz, Austria; ahmed.mahdy@stud.medunigraz.at (A.M.H.); andreas.roessler@medunigraz.at (A.R.); 2Internal Medicine Department, Faculty of Medicine, Ain Shams University, Ramsis Street, Abbassia Square, Cairo 11435, Egypt; 3Rheumatology and Immunology Department, Medical University of Graz, Auenbruggerplatz 15, 8036 Graz, Austria; martin.stradner@medunigraz.at; 4Department of Health Sciences, Alma Mater Europaea, Slovenska ulica 17, 2000 Maribor, Slovenia

**Keywords:** atherosclerosis, autoimmune disease, endothelial dysfunction, oxidized LDL, antioxidized LDL antibodies, nitric oxide

## Abstract

Autoimmune rheumatic diseases (AIRDs) with unknown etiology are increasing in incidence and prevalence. Up to 5% of the population is affected. AIRDs include rheumatoid arthritis, system lupus erythematosus, systemic sclerosis, and Sjögren’s syndrome. In patients with autoimmune diseases, the immune system attacks structures of its own body, leading to widespread tissue and organ damage, which, in turn, is associated with increased morbidity and mortality. One third of the mortality associated with autoimmune diseases is due to cardiovascular diseases. Atherosclerosis is considered the main underlying cause of cardiovascular diseases. Currently, because of finding macrophages and lymphocytes at the atheroma, atherosclerosis is considered a chronic immune-inflammatory disease. In active inflammation, the liberation of inflammatory mediators such as tumor necrotic factor alpha (TNFa), interleukine-6 (IL-6), IL-1 and other factors like T and B cells, play a major role in the atheroma formation. In addition, antioxidized, low-density lipoprotein (LDL) antibodies, antinuclear antibodies (ANA), and rheumatoid factor (RF) are higher in the atherosclerotic patients. Traditional risk factors like gender, age, hypercholesterolemia, smoking, diabetes mellitus, and hypertension, however, do not alone explain the risk of atherosclerosis present in autoimmune diseases. This review examines the role of chronic inflammation in the etiology—and progression—of atherosclerosis in autoimmune rheumatic diseases. In addition, discussed here in detail are the possible effects of autoimmune rheumatic diseases that can affect vascular function. We present here the current findings from studies that assessed vascular function changes using state-of-the-art techniques and innovative endothelial function biomarkers.

## 1. Introduction

Until now, no evidence about the true mechanism of the atherosclerosis process in autoimmune rheumatic diseases (AIRDs) is known [1]. Current research findings are often contradictory. Whether traditional risk factor like age, gender, smoking, or hypertension contribute solely to the atherosclerosis, or whether it is a consequence of a nontraditional states like chronic inflammation or changes in cytokines, antibodies that accompany atherosclerosis, or a combination of both aspects is not yet clear and is debated [1]. Moreover, it is currently not known which factors play a major role in process of development of premature atherosclerosis in autoimmune rheumatic disease patients.

Atherosclerosis can be classified into primary simple atherosclerosis, which occurs with age, and secondary autoimmune atherosclerosis, which was also coined as accelerated atherosclerosis [2]. Atherosclerosis is the main cause of cardiovascular diseases in autoimmune rheumatic diseases. This review examines the role of chronic inflammation in the etiology—and progression—of atherosclerosis in autoimmune rheumatic diseases. In addition, discussed here in detail are the possible effects autoimmune rheumatic diseases can have on vascular function, as well as how their effects on vascular function could potentially be assessed using state-of-the-art techniques and innovative biomarkers.

## 2. Methodology

The current literature on the subject of “autoimmune disease and vascular function” was systematically reviewed. Both primary and secondary sources of literature on the topic were examined. PubMed and Web of Science were used as search engines to access the relevant literature. The initial search criteria included the following keywords: autoimmune atherosclerosis, connective tissue autoimmune diseases and atherosclerosis, premature atherosclerosis in autoimmune diseases, vascular dysfunction, vascular dysfunction in autoimmune diseases, rheumatoid arthritis and cardiovascular diseases, rheumatoid arthritis and atherosclerosis, systemic lupus erythematosus and cardiovascular diseases, lupus and atherosclerosis and premature atherosclerosis, Sjögren’s syndrome and cardiovascular diseases, Sjögren’s syndrome and atherosclerosis, mortality and morbidity in autoimmune diseases, mechanism of atherosclerosis in autoimmune diseases, measurement of endothelial function, endothelial dysfunction, vasculitis, autoimmune vasculitis, atherosclerosis as immune diseases, new theory of atherosclerosis, blood marker of endothelial dysfunction, oxidized LDL anti oxidized LDL antibodies, amyloidosis in autoimmune diseases, traditional risk factor of atherosclerosis, nontraditional risk factor in atherosclerosis, Cryoglobulinemia and Cryoglobulinemic vasculitis, Cryoglobulin and autoimmune diseases. The search resulted in 16,000 articles from PubMed and Web of Science.

After exclusion of animal studies and duplicate studies and studies in languages other than English language, and after restricting the search to only the last 5 years (2016–2020), 500 relevant papers were identified. Next, articles that did not fulfil our research concern were removed. This resulted in 180 papers. Finally, the reference lists of the selected papers were examined for any relevant papers (Figure 1).

## 3. Autoimmune Rheumatic Diseases (AIRD)

Autoimmune rheumatic diseases are diseases of unknown etiology, but genetic, environmental, food, and stress are considered risk factors. It affects about 5% of the general population [3]. Currently, the incidence of most autoimmune rheumatic diseases has been increasing; no specific causes were, however, attributed to their current increase [4]. Lack of clear etiology made autoimmune diseases a challenging health problem, as they are difficult to diagnose and manage [5]. When the body starts to lose its tolerance to self-antigens and considers its own tissues and organs as an enemy, it begins to attack the tissues, leading to marked deterioration of the organs’ functions, as well as quality of life [6]. The mortality rate of AIRDs is high, especially in Systemic lupus Erythematosus (SLE) patients [7]. The causes of mortality in autoimmune diseases are mostly active diseases, infections, and/or cardiovascular diseases [8,9,10]. In addition, the morbidity and mortality associated with cardiovascular diseases in these patients may be doubled or tripled as compared to that of the normal population [11]. Indeed, up to 30–50% of the mortality in rheumatological autoimmune diseases arises due to cardiovascular diseases [12,13]. Chronic systemic inflammation, which occurs in autoimmune diseases, and the accompanying release of cytokines accelerate the atherosclerosis process either via unknown causes [14,15,16], the disease itself and/or a consequence of the disease’s treatment, or due to their complications. With every relapse, and recurrent acute inflammation, the prognosis of the disease becomes much worse, especially with respect to the occurrence of cardiovascular events [3].

### Pathophysiology of Autoimmune Rheumatic Diseases

The connective tissue is the tissue that connects the organs of the body and connects the parenchyma and stroma of the organs. It consists mainly of elastin and collagen. Collagen fibers are present in blood vessels, eyes, bones, joints, and skin. Inflammation of connective tissues leads to organ dysfunction. In normal conditions, the function of the immune system of the body is to defend against infection and to aid repair of tissue injury. Self-antigen is not recognized by the immune system as foreign. In AIRDs the immune system loses its tolerance to certain self-antigens, starting to attack it [6]. This attack includes release of proinflammatory cytokines, direct cytotoxicity, and production of antigen-specific antibodies, leading to complement fixation and tissue inflammation, and potentially resulting in organ damage and failure [17].

The causes of AIRDs are unknown, but genetic factors are believed to be implicated in the susceptibility of some persons to develop autoimmune diseases. Similarly, exposure to environmental factors like toxins, microbes, food, and stress may initiate the development in the future of AIRDs [18].

## 4. The Effect of AIRDs on Vascular Function (Premature Atherosclerosis)

Figure 2 below provides an overview of the traditional and nontraditional factors that play important roles in the development of atherosclerosis and/or hypertension in auto-immune rheumatic diseases. These are discussed in detail below.

### 4.1. Traditional Risk Factors in Atherosclerosis

Cardiovascular diseases are considered as one of the most common causes of morbidity and mortality in autoimmune rheumatic diseases [19]. Recent studies, however, show contradictory results, and there is currently a debate about the role of traditional and nontraditional risks in the development of atherosclerosis processes, which, in turn, are believed to play important role in the development of cardiovascular diseases in autoimmune rheumatic diseases [12,20]. Traditional risk factors include age, gender, smoking, obesity, dyslipidemia, hypertension, diabetes, and a sedentary lifestyle [12,21]. Interestingly, there is some controversy about body weight in relation to disease activity, especially in rheumatoid arthritis. Some studies named it “inflammatory cachexia” [22], in active inflammation, the liberation of inflammatory mediators, such as TNFa, IL6, IL1, and others that affect the body metabolism, leads to a state of hypercatabolism. This hypercatabolism is in addition to the anorexia associated with acute inflammation in these patients, thus leading to decreases in body weight (mostly in muscle mass) rather than obesity [23,24]. However, most studies show an increased prevalence of obesity in RA patients [25,26,27]. Other studies reported that even though obesity is a contributing risk factor in development of atherosclerosis process, obesity has an otherwise good prognostic and protective effect in cardiovascular mortality in comparison to that of low body weight in rheumatoid patients. For instance, a BMI more than 30 was reported to have a better prognostic outcome as compared to that of a BMI of less than 20 [28]. 

Similarly, “Lipid paradox” was also reported in patients with autoimmune rheumatic diseases. “Lipid Paradox” is a term applied for lipid profile in AIRDs, and perhaps even in all inflammatory conditions, where, during the active inflammatory state, there is a decrease in all lipid profiles: total cholesterol, LDL, and HDL cholesterol (that is, there is a state of hypercatabolism). However, once the active inflammatory subsides (spontaneously or with drugs), there is a state of normal or increased serum cholesterol levels [29].

Smoking is a well-known independent risk factor for atherosclerosis [30]. In addition, most studies show that it increases the risk of cardiovascular diseases in RA patients [31], especially if smoking is accompanied by the presence of rheumatoid factor (RF) and/or anticitrullinated peptide antibody (ACPA) [32]. Hyperhomocysteinemia, known as a traditional risk factor in cardiovascular diseases [33], is also present in AIRDs. It is considered one of the independent risk factors in the severity of the autoimmune diseases [34]. Although there is debate about obesity and cachexia in AIRDs, all studies suggest that metabolic syndrome is associated with disease severity and increased atherosclerosis risk in autoimmune diseases [29,35]. Hyperuricemia is considered a risk factor in cardiovascular diseases, and also in AIRDs [36]. Finally, insulin resistance was also suggested to play a role in autoimmune rheumatic diseases [29].

### 4.2. Nontraditional Risk Factors in Atherosclerosis

Nontraditional risk factors for atherosclerosis refer to those risk factors that are not related to the usual Framingham cardiovascular risk factors, such as age, gender, etc., and are specific to certain diseases. These classically include cells, cytokines, chemokines, antibodies, and drugs (see Figure 2). Endothelial cells have a major role in the state of endothelial dysfunction. During inflammation, there is a release of many cytokines, which influence the action of nitric oxide and lead to vasoconstriction, as well as increase the risk of thrombosis and atheroma formation [37]. Also, the cells of the immune system especially T lymphocytes and B lymphocytes are implicated in this process. For instance, T cells are involved in the processes of atherosclerosis, and are a major contributor to AIRDs [38]. T helper cells, whether T helper 1 [39] or T helper 17 (Th17) cells, have a stimulatory effect on atherosclerosis formation [14]. On the other hand, regulatory T cells (Treg) have a protective function on the atherosclerosis process [40]. Unsurprisingly, the ratio of helper Th17 cells to regulator T reg cells is important in atherosclerosis formation and in controlling the disease during therapeutical interventions (e.g., with medications) [39]. With regards to immune mediators, TNFa is one of the most important mediators in autoimmune atherosclerosis and autoimmune diseases. TNFa is secreted from macrophages, endothelial cells, and T lymphocytes. As TNFa initiates or potentiates acute inflammatory states, anti-TNFa could play a major role in the control of the inflammatory conditions and disease control [41]. IL6 is also considered an important cytokine in inflammatory conditions. IL6 is found in RA patients with premature atherosclerosis [42]. Furthermore, IL1 was reported to have a role in the development of atherosclerosis in AIRDs [43].

Antibodies are basic contributors in the pathogenesis of autoimmune diseases. Many antibodies were reported to be implicated in the development of autoimmune rheumatic diseases and autoimmune atherosclerosis [44]. Rheumatoid factor (RF) is an antibody associated with rheumatoid arthritis, and its levels correlate with arthritis activity and prognosis; its presence is also associated with increase in cardiovascular risk [45,46]. Also, anticitrullinated protein antibody (ACPA) is associated with aggressive rheumatoid diseases; their presence is associated with worsening of the prognosis in RA and greater cardiovascular manifestations [47]. Antinuclear antibodies (ANA) are secreted in many AIRDs, and their levels are also related to diseases activity and formation of atherosclerotic plaques [48]. Anti-DNA antibody, which is specific to systemic lupus patients, also relates to disease activity [49]. Anti-LDL antibodies are antibodies formed against oxidized LDL, and its presence in the blood of the patients indicates an increased incidence of vascular dysfunction and cardiovascular deconditioning [50]. However, other studies indicate that antioxidized LDL confers protection against cardiovascular risk [51,52]. Interestingly, studies on the effect of anti-LDL antibodies showed that IgM is protective against atherosclerosis, while IgG has harmful effects [50]. The presence of antiphospholipid antibodies (e.g., in antiphospholipid antibody syndrome or its presence as secondary antiphospholipid in other connective tissue diseases) indicates a major risk for development of vascular thrombosis and cardiovascular diseases [53]. Cryoglobulin, immunoglobulins that precipitate in vitro at lower temperatures, was shown to be associated with many autoimmune diseases, especially Sjögren’s syndrome, systemic lupus, and hepatitis C viral infection [54]. Its presence in the blood does not indicate the presence of disease, but it can cause many vascular symptoms due to its precipitation and obstruction of blood vessels of multiple organs ranging from critical vascular ischemia to autoimmune vasculitis [55], which aggravate endothelial dysfunction [56]. Although anti-Ro anti-La antibodies are associated with congenital heart block in neonatal lupus disease [57], the presence of anti-Ro antibodies in adults may be associated with complete heart block in primary Sjögren’s syndrome patients [58,59].

### 4.3. Cardiac Amyloidosis

Normal cardiac output heart failure is the type of heart failure that is seen in rheumatoid arthritis patients [60]. Previously, researchers attributed heart failure in RA to ischemia of coronary arteries, but subsequently, heart failure in these patients was seen with preserved systolic function, which was not explained by coronary ischemia [60,61]. Others reported that heart failure may arise due to deposition of protein fibril between cardiac muscle; this protein material deposit in the heart occurs due to a chronic inflammatory state [62]. This refers to the possibility of amyloid deposits in cardiac muscle (secondary Amyloidosis, AA). Amyloidosis is classified as primary amyloidosis (AL), which is caused by hematological diseases such as multiple myeloma and secondary amyloidosis, which is a result of chronic inflammation a state and deposition of serum amyloid A in a chronic inflammatory situation like immune diseases or chronic infection [63]. Primary amyloidosis affects mainly the heart but secondary amyloidosis, although it affects mainly the kidney leading to proteinuria and nephrotic syndrome, can affect any organ in the body including the heart [64,65]. As secondary amyloidosis results from a chronic inflammatory state, and most AIRDs persist for more than 10 years, we speculate that cardiac amyloidosis could have led to heart failure in AIRDs. This could lead to increased mortality in these patients.

### 4.4. Hypertension in Autoimmune Rheumatic Diseases 

The prevalence of hypertension in autoimmune rheumatic diseases is high. Indeed, there are several studies that showed that hypertension is very common in rheumatoid arthritis patients [66]. There is a wide range of prevalence of hypertension in these studies [67,68,69,70,71]: with some studies reporting hypertension in rheumatoid patients between 3–70% [72]. However, other studies reported no differences in blood pressure between patients and controls [73,74,75,76,77,78]. Therefore, there is still a debate about the percentage of hypertension in autoimmune diseases [3,70,71]. However, some study-related hypertension in rheumatoid arthritis was seen when age and disease duration of the patients are considered [79]. As hypertension and autoimmune rheumatic diseases can be influenced by genetic and environmental factors, it is often difficult to identify which of these started first. For instance, hypertension leads to mechanical injury of the endothelium, which then starts the sequence of endothelial dysfunction, loss of endothelial vasodilation substances, arterial stiffness, and atherosclerosis, which is most probably helped by the state of chronic inflammation in autoimmune patients. On the other hand, in chronic autoimmune rheumatic diseases, endothelial injury arises due to chronic inflammation, antigen–antibody reaction, complement fixation, and/or disturbance in the normal equilibrium of cytokines and chemical mediators, which can lead to vasoconstriction (especially in renal vasculature), and increase vascular resistance and subsequent hypertension [80]. Roman and colleagues illustrated that there is no difference regarding hypertension prevalence between the control group and rheumatoid group [77].

In general, hypertension is the single most important cardiovascular risk factor that can accelerate the development of premature atherosclerosis, and consequently, cardiovascular diseases [21]. Overall, age and hypertension are the main factors responsible for arterial stiffness; both are considered as the most important factors responsible for atherosclerosis and development of subsequent cardiovascular diseases [80]. The combination of autoimmune rheumatic diseases and hypertension is more dangerous than autoimmune rheumatic diseases alone [81]. For example, the treatment of autoimmune diseases with drugs, especially corticosteroids, could potentially influence hypertension control in these patients [81]. All studies from systemic lupus erythematosus patients show that the prevalence of hypertension is very high in SLE, especially when the kidneys are also affected. Renal diseases (both glomerulonephritis or tubular diseases) lead to elevation of blood pressure via salt-water retention and/or ischemia to the glomerulus and consequent activation of the renin-angiotensin system [82,83,84,85,86].

Hypertension is now considered an immune disease, as data from experimental animals show that the deletion of B cells by anti-CD 20 influences blood pressure control [87,88]. Also, the change in the ratio of helper T cells 1 and 17 and T regulator cells modulates the control of blood pressure [89,90]. The problem of hypertension in autoimmune diseases is that the true mechanism of hypertension development is not known; no guidelines for hypertension control in autoimmune diseases are available, thus making the control of hypertension in autoimmune diseases very difficult [81]. Furthermore, the effect of corticosteroid therapy on hypertension is still unclear as it produces salt and water retention, thus leading to increases in blood pressure on the one hand, but as its anti-inflammatory activity decreases, cytokine production consequently decreases blood pressure on the other hand [72,91]. Most of the antihypertensive medications show little effects on control of the blood pressure in autoimmune rheumatic diseases. However, angiotensin converting enzyme inhibitors (ACEIs) and selective angiotensin receptor blockers (ARBS) are effective in blood pressure regulation in these patients as they have both immunomodulatory and antihypertensive effects [81,92].

### 4.5. Rheumatoid Arthritis

Cardiovascular signs in rheumatoid arthritis—due to the accompanying accelerated atherosclerosis—present earlier in the course of the disease. These signs may sometimes appear before even a diagnosis is established [93]. Also, the number of rheumatoid arthritis patients with cardiovascular manifestations can double within one-year of follow-up [94]. Common classical (“traditional”) risk factors related to cardiovascular disease in RA include male sex, age, hypertension, hypercholesterolemia, smoking, and obesity. However, the increased incidence of vascular dysfunction, premature atherosclerosis, and cardiovascular diseases in rheumatoid arthritis is not fully explained by these traditional factors [95]. Overall, up to 70% of cases of RA can be attributed to classical risk factors, but the remaining 30% of RA patients have no known relevant risk factors associated with development of cardiovascular diseases [96,97]. It is possible that a combination of nonclassical risk factors and classical risk factors predispose RA patients towards development of cardiovascular diseases. Or perhaps there remains another risk factor that is not yet identified [98]. Ischemic heart diseases—arising due to atherosclerosis and narrowing of coronary vessels—are considered common presentations, as well as leading cause of death, in rheumatoid arthritis patients. The risk of myocardial infarction in rheumatoid patients is similar to that in diabetic patients [99].

Humphreys and colleagues (2014) suggested that, despite current advances in diagnosis and management of rheumatoid arthritis, including biological therapy, the mortality rate did not fall within the last 20 years [100]. On the other hand, Myasoedova and colleagues (2017) reported that in the last 20 years the mortality of rheumatoid arthritis decreased when compared to that of the normal population. They suggested that this decrease in mortality possibly arose due to the effects of modulatory drugs (e.g., traditional immunomodulatory or biological therapy) [101]. The average life expectancy of patients with rheumatoid arthritis can decrease by up to 10 years when compared to that of the normal population [102]. In fact, cardiovascular disease is considered a major cause of death in rheumatoid arthritis patients worldwide [103]: one-third or more of mortality in RA is due to cardiovascular disease. There is also a 3-fold mortality rate in RA patients as compared to that of the normal population [104]. As compared to that of the normal population, there is a doubling of sudden cardiac death in rheumatoid arthritis [48].

Senescent CD4^+^ CD28^−^ T helper cells play a role in the pathogenesis of atherosclerosis in rheumatoid patients. The patients who increased amount of CD4^+^ CD28^−^ T helper cells have higher probability of developing increased intima-media thickness [44]. Also, the level of RF and ACPP are correlated with cardiovascular risk [45,46].

All studies in which the effect of rheumatoid arthritis on flow-mediated dilatation was investigated show impairments in brachial artery reactive hyperemia [105,106]. In addition, increases in arterial stiffness and carotid intima-media thickness were also seen thus suggesting premature atherosclerosis [107,108,109,110,111] in these patients. Many of the biological drugs used in RA antagonize the effect of inflammatory cytokines like TNF alpha or IL-6. These antagonists of the inflammatory cascade improve flow-mediated dilation, and subsequently, decrease atherosclerosis and cardiovascular events [111].

There seems to be no correlation between macrovascular and microvascular endothelial function in rheumatoid arthritis (that is, they appear to be independent of each other) [112].

### 4.6. Systemic Lupus Erythematosus

In SLE, there is a loss of immune tolerance and spontaneous activation of the immune system with the production of the autoantibody against nuclear proteins, termed antinuclear antibodies (ANA). Deposition of antigen-antibody immune complexes in blood vessels leads to organ injury [113]. SLE affects women predominantly, with a proportion of female-to-male 12:1 during the childbearing period (this ratio decreases as females age). The disease can start at any age of life, but early diagnosis is associated with poor prognosis. In organs such as the heart, brain, and kidney, failure is associated with high mortality. Vasculitis, a common presentation in lupus patients, arises when direct antibodies against endothelial cells as well antigen-antibodies complex become deposited in blood vessels [114,115]; it could also arise due to vasculitis in situ. Vasculitis aggravates the state of endothelial dysfunction with the rapid development of premature atherosclerosis [12,44].

Besides the common traditional risk factors associated with atherosclerosis, other associated factors related to lupus disease itself appear to be more important towards the aggravation of atherosclerosis [20,116]. It is related to chronic inflammation and immune complex deposition as well as the disease or therapy related complications. Zhu and his colleagues found that the serum level of IL-10 in SLE patients with atherosclerosis was less than in that of the SLE patients without atherosclerosis. Also, they found elevated levels of serum IL-6 and IL-17 in the atherosclerotic group. In addition, they reported that the amount of regulatory T cells (Treg) in atherosclerotic SLE was lower than those found in SLE patients without atherosclerosis. On the other hand, the number of T helper 17 cells (Th17) in atherosclerotic SLE group was higher as compared to that of patients without atherosclerosis [40].

Drugs used in the treatment of systemic lupus erythematosus and other autoimmune diseases such as corticosteroids in high doses could lead to hypertension and diabetes mellitus [117]. On the other hand, corticosteroids in low doses (acting as anti-inflammatory agents) can prevent atherosclerosis [118]. Other immunosuppressive drugs such as hydroxyl-chloroquine, which may be protective on vessels as potent immunomodulatory agents, and drugs used like nonsteroidal anti-inflammatory drugs, especially anti cox-2 used as analgesics, could have side effects on cardiac muscle functions.

Coronary arterial disease is a major cardiovascular risk in SLE [119]. Women aged 35–44 years are more likely to have a high risk of developing myocardial infarction and acute coronary syndrome, with an increase in mortality, as compared to that of women of a similar age [11,20]. In general, the risk of developing myocardial infarction in SLE is doubled as compared to the normal population [119]. Interestingly, in women aged 40–49 having SLE, the risk of myocardial infarction increases by 8-times during a 7-year follow-up [115,120]. Females with SLE below age 45 years have an increased incidence of cardiovascular disease as compared to that of same-age women. These diseases include hypertension, especially with involvement of the kidneys [121,122], pulmonary hypertension, myocarditis, and stroke [123]. 

Cardiovascular disease risk in lupus patients is two-fold as compared to that of the normal population [124]. Classically, there are two peaks in SLE mortality, with the first peak appearing in the first 3 years of the disease diagnosis. This peak is due to activity of the disease, the complication of immunosuppressant drugs decreasing immunity, increased infections, and kidney involvement. The second peak of mortality ends with death, occurring after a long period lasting from 4–20 years following diagnosis and treatment, resulting mainly from cardiovascular diseases [20,116]. Aviña–Zubieta and his colleagues (2017) reported that the mortality from cardiovascular disease and myocardial infarction in the first year of diagnosis is possible [120].

FMD is impaired in lupus erythematosus patients; interestingly, the impairment is mostly found in older age patients with hypertension, diabetes, and renal impairments [125,126]. Arterial stiffness is increased in lupus patients with more relation to age, hypertension, and glucocorticoid use [127,128,129]. Carotid plaque is 21% in patients with lupus under 35 years, and the percentage increased and reached 100% for women over age 65 years [20,130], which may be related to age or disease activity or both.

### 4.7. Primary Sjögren’s Syndrome(PSS)

PSS is an autoimmune rheumatic disease that affects mostly women above 40 years. It is characterized by lymphocytic infiltration, chronic inflammation, and destruction of the salivary and lacrimal glands, leading to xerostomia and xerophthalmia, respectively [131]. PSS can also affect other extra glandular organs, such as blood vessels, thyroid, joint, kidney, and lung [132]. Although studies related to the investigation of cardiovascular risk with PSS are limited, PSS is associated with some degree of vascular dysfunction, rigidness of blood vessels, and arterial stiffness that can lead to subclinical atherosclerosis. However, these cardiovascular effects of PSS cannot be explained by traditional risk factors alone [133]. Demirci et al., (2016) suggest that the increase in vascular stiffness in PSS may not be related to the disease per se but rather to the usage of corticosteroids, accompanying hypertension, and/or abnormal lipid profile, which may accompany the disease [134]. 

Chronic inflammation, cytokine production, and antibodies present in PSS patients could all contribute towards vascular dysfunction [135]. Antioxidized LDL antibodies, which are atheroprotective in nature, are low in PSS. This finding in PSS patients is in contrast to what is seen in SLE patients, indicating that the antibodies have a role in atherosclerosis formation [51].

For several years, lymphoma was considered the leading cause of mortality and morbidity in PSS patients, but currently, cardiovascular diseases are identified as the major causes for mortality in PSS patients [136]. 

Overall, PSS appears to be associated with some degree of vascular dysfunction, rigidness of blood vessels, and arterial stiffness that can lead to subclinical atherosclerosis [133,134]. 

Thus, it is expected that drugs that interfere with inflammatory processes can reduce inflammation and subsequently reduce disease activity and the development of atherosclerosis [111].

Table 1 below provides an outline of different studies that examined the effects of autoimmune rheumatic diseases on vascular function and assessed the risk of accompanying atherosclerosis and the development of cardiovascular diseases.

Dysfunction of the endothelium initiates vascular injury and commences the process of atherosclerosis. The following are the physiological effects of nitric oxide (NO): It keeps the tone of vessels, ensures laminar blood flow, prevents leucocyte migration and adhesion, plays important role in the maintenance of homeostasis, and prevents platelet aggregation, and, consequently, prevents atherosclerosis [137,138,139].

Asymmetric dimethylarginine (ADMA), an analogue of L-arginine, is a naturally occurring product of metabolism found in human circulation. ADMA is known to antagonize the action of nitric oxide on blood vessels via inhibition of the enzyme nitric oxide synthase. Hence, ADMA contributes towards dysfunction of endothelium, and consecutively, vessel diseases [140,141]. Current literature indicates that endothelial dysfunction is an early sign of atherosclerosis and all cardiovascular events and future diseases [142,143].

Defects in nitric oxide synthesis, release or its function are often associated with the commencement of endothelial dysfunction and atherosclerosis. Nitric oxide is synthesized in the endothelial cell layer by enzyme nitric oxide synthase (NOS) from amino acid arginine. ADMA is a competitive inhibitor of NOS that leads to a decrease in nitric oxide formation [144,145].

Atherosclerosis was previously considered a degenerative disease and was always associated with the aging process. However, new research has shown that atherosclerosis is not a disease of only older persons and is also not an inevitable disease. There are immune and inflammatory factors associated with its pathogenesis and lipoprotein metabolism, which play an important role in atheroma formation through activation of the immune system [16,146] and participation of oxidized LDL and antioxidized LDL [147,148,149,150,151]. The innate immune mechanisms were shared in the initiation of vascular dysfunction, hypertrophy of smooth muscle cells layer, atheroma formation, and arterial narrowing through the production of cytokines and inflammatory mediators [152]. Also, the adaptive immune system through B and T cell lymphocytes [16,152], whether T helper (T4) or T cytotoxic (T8) [153], are found in plaque section with antibodies [154]. Atherosclerosis consists of fatty degeneration, which is responsible for arthrosis with plaque formation, and vessel stiffening, which refers to sclerosis. Plaque is formed of the precipitation of lipoprotein, macrophage (foam cells), antibodies, and cytokines within the vessel wall [155]. All these factors are responsible for the gradual thickening of the intima and media with stiffness of the vessel wall, resulting in rigidity and narrower vessels with turbulence blood flow, rupture of the plaque, and cardiovascular complications. Figure 3 below provides an overview of the possible underlying mechanisms of immune-atherosclerosis.

## 5. Measurements (Assessment) of Vascular Function

The measurements of endothelial dysfunction (vascular function) are important, as they are considered early predictors of cardiovascular risk, future morbidity, and mortality [143]. Mortality can be increased due to cardiovascular diseases like myocardial infarction and cerebrovascular stroke [143]. Assessment of vascular function can be done by showing the arteries themselves directly, which means we measure its direct function or can be indirect with measuring blood marker, which denotes their function. Both atherosis and sclerosis processes can be measured with noninvasive ultrasound devices. In atherosclerosis in the arteries, for example, carotid artery can be measured with ultrasound, and plaque and thickness of intima-media assessed [156]. Arterial plaque and intima-media reflect the anatomical structure, while stiffness reflects the function of the arteries. All these are subclinical detectors for cardiovascular disease and a good way to predict future cardiovascular events [157,158]. 

There are several techniques that can be used to assess vascular function and endothelial health [159]. Figure 4 below provides an overview of these selected but important measurements of vascular function assessments, including some biomarkers, that are widely used in the literature.

Some of these assessment techniques of vascular function are detailed below:

### 5.1. Flow-Mediated Dilation (FMD)

Endothelial (dys) function can be assessed using flow-mediated dilatation. FMD involves ultrasonic assessment of the percentage increase in brachial artery diameter from baseline conditions to maximum vessel diameter during hyperemia induced by inflation and deflation of a sphygmomanometer cuff to supra systolic levels for 5 min [160,161,162]. The principle of this technique is that when there is a cutting off blood supply to the tissues, there is a collection of metabolites (NO, for example) that lead to compensatory vasodilation after the return of circulation. In normal healthy endothelium, occlusion of the artery liberates these metabolites, which, in turn, leads to dilation of the vessel. In unhealthy persons, who often have low levels of NO, no increases in vascular diameter are seen when the arterial occlusion is released. The gold standard of noninvasive endothelial function testing is FMD of the brachial artery [163,164,165]. As the FMD measurements are operator-dependent, strict physiological and methodological should be used [166,167,168,169].

### 5.2. Pulse Wave Velocity (PWV)

Vascular stiffness of big blood vessels measured using pulse-wave velocity which detects the rate of arterial pulse waves move along the vessel wall [170]. The Vicorder instrument (SMT medical GmbH & Co. KG, Würzburg, Germany) used to investigate changes in arterial stiffness during rest. PWV needs two-point to measure the speed of wave transmission between them through the arterial system, and the measure between carotid and femoral is the best. For the recording, the subject needs to lie down in the bed. One cuff has to be fixed around the neck so that the probe lies above the carotid. The second cuff should be fixed as high as possible around the right thigh for femoral artery detection [171].

### 5.3. Carotid Intima-Media Thickness

The measure of intima-media thickness is an easy way to find endothelial dysfunction [172]. As in normal endothelium, there is no plaque, and the average thickness on intima-media is 0.9 mm [164], but when the endothelium does not function well and becomes unhealthy, there is an increase in its thickness, and atheromatous plaque forms [173]. Carotid or femoral artery can easily accept big arteries for detecting either plaque or thickness of intima-media by ultrasound device.

The call-out Box 1 below summarizes the current literature related to assessment of FMD, pulse wave velocity, and carotid intima media thickness:

Box 1Summary of the findings related to assessment of vascular function and atherosclerosis in different autoimmune rheumatic diseases.*Rheumatoid arthritis*: All studies regarding the effect of rheumatoid arthritis on flow-mediated dilatation showing impairment in its reactive hyperemia [105,106] and increase in arterial stiffness and carotid intima media thickness showing the evidence of premature atherosclerosis [107,108,109,110,111]. There is no correlation between macrovascular and microvascular endothelial function in rheumatoid arthritis (independent of each other) [112].*Systemic Lupus Erythematosus*: FMD is impaired in lupus erythematosus patients; interestingly, the impairment is mostly found in older age patients with hypertension, diabetes, and renal impairments [125,126]. Arterial stiffness is increased in lupus patients with more relation to age, hypertension, and glucocorticoid use [127,128,129]. Carotid plaque is 21% in patients with lupus under 35 years, and the percentage increased and reached 100% for women over age 65 years [20,130], which may be related to age or disease activity or both.*Primary Sjögren’s Syndrome (PSS)*: PSS appears to be associated with some degree of vascular dysfunction, rigidness of blood vessels, and arterial stiffness, which can lead to subclinical atherosclerosis [133,134].

### 5.4. Retinal Imaging

Another noninvasive technique involving retinal microvasculature assessment can also be used to assess vascular function. Retinal microvasculature showed that venules are not only passive vessels, but they represent the state of dynamic components responsive to changes in the microcirculation [174]. Other recent studies reported that dilated retinal venules are denoting inflammation, endothelial dysfunction, and cerebral hypoxia. On the other side, constricted retinal arterioles correspond to endothelial dysfunction and elevated blood pressure [175]. It was reported that aging affects the microvasculature of the retina [176].

Digital retinal camera scans can be proposed for noninvasive retinal vessels’ assessment. The portable or table mounted retinal cameras are designed for a comfortable, painless, and quick examination of the retina. Retinal images are stored on a laptop and data analysis is performed. This analysis is done using a commercially available retinal imaging and analysis software. During analysis of the retinal microvascular, the largest six arterioles and venules appearing through a zone between half and one-disc diameter from the optic disc margin are measured [177]. The measurements are described as central arteriolar equivalent (CRAE), which denotes the diameter of retinal arteries and central retinal venular equivalent (CRVE), which denotes diameter of retinal veins. The ratio between the CRAE and CRVE is also measured.

### 5.5. Endo-Pat Test (Peripheral Arterial Tonometry)

This test assesses endothelial function. It depends on peripheral arterial tone (PAT) signal, which is measured from the fingertip by measuring arterial pulsatile volume changes [178]. It is a reliable, objective, and not dependent on human factors (e.g., skill of the ultrasonographer). Furthermore, the derivation of the reactive hyperemia index (RHI) is automated.

### 5.6. Blood Markers

There are several blood markers that can detect the function of the endothelium directly or indirectly (Figure 4). Oxidized LDL can be measured in the blood. Similarly, antibodies to oxidized LDL, which may be responsible for the protection or harming endothelium during the process of atherosclerosis (wither IgM or IgG), can be measured. Therefore, antibodies to oxidized LDL were introduced as new biomarker for assessment of endothelial health [148,149,150,151]. Interestingly, high-density lipoprotein (HDL), which is anti-atherosclerosis and a scavenger for LDL in the blood, when LDL becomes oxidized, HDL lost its function and converted to proinflammatory HDL (piHDL) [179]. Also, ADMA, which is considered the most important indicator for endothelial dysfunction, can be used as a biomarker for endothelial health assessment. Cryoglobulin is a marker for evaluation of blood vessel vasculitis, and it is a good indicator, especially in Sjögren’s syndrome, to detect vasculitis and malignant lymphoma transformation risk [180]. Similarly, antiphospholipid antibodies are very important markers of vascular function, especially in patients with antiphospholipid syndrome [181]. Additionally, pentraxin-3 is an inflammatory marker that increases in cardiovascular diseases and AIRDs [182]. Finally, Endocan is another mediator of endothelial dysfunction and is related to the development/monitoring of atherosclerosis and cardiovascular diseases [183,184].

## 6. Conclusions

Mortality can increase in autoimmune rheumatic diseases due to cardiovascular diseases, especially myocardial infarction and cerebrovascular stroke. Therefore, there is a need to diagnose early on the state of accelerated atherosclerosis in autoimmune diseases. This review provides an overview of selected autoimmune rheumatic diseases and discusses the roles of traditional and nontraditional factors in the development of cardiovascular diseases in these patients. Comprehensive studies including state of the art vascular function measurements in different population groups and stages of the diseases are required in the future.

## Figures and Tables

**Figure 1 jcm-10-04427-f001:**
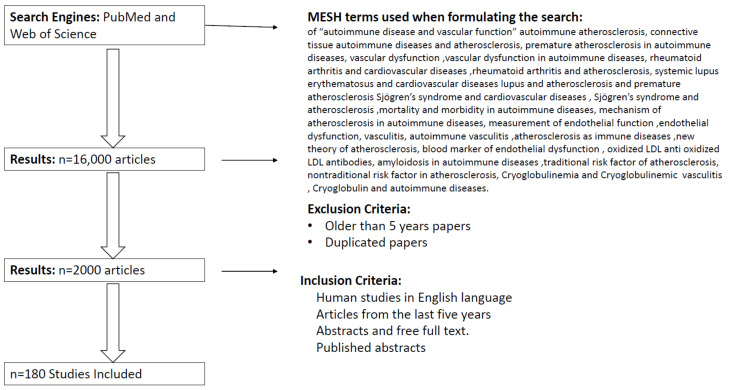
Overview of studies selection.

**Figure 2 jcm-10-04427-f002:**
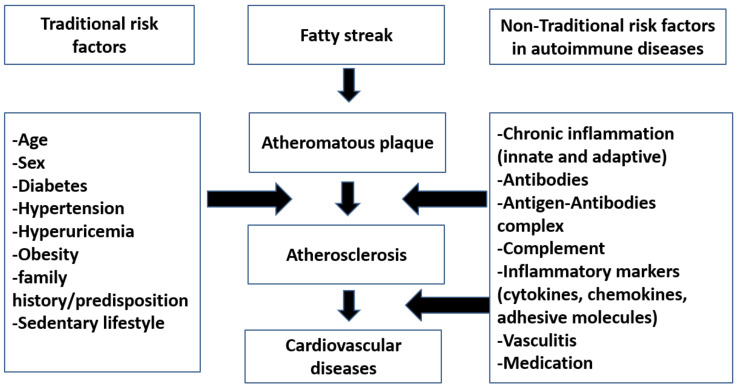
An overview of roles of traditional versus nontraditional risk factors in atherosclerosis in autoimmune diseases.

**Figure 3 jcm-10-04427-f003:**
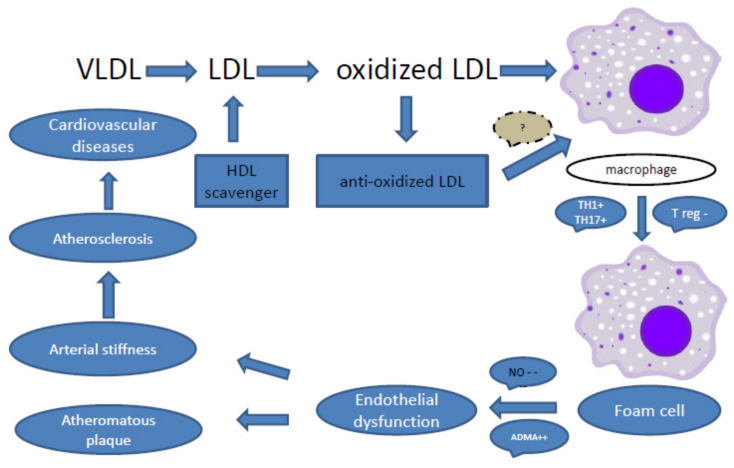
Pathogenesis of atherosclerosis based on possible association with immune changes accompanying inflammation. Legend: VLDL: very-low-density lipoprotein; LDL: low-density lipoprotein; HDL: high-density lipoprotein; TH1: T helper cell 1; TH17: T helper 17; T reg: T regulator cells; NO: nitric oxide; ADMA: asymmetric dimethylarginine; ++ stimulate; −− inhibit.

**Figure 4 jcm-10-04427-f004:**
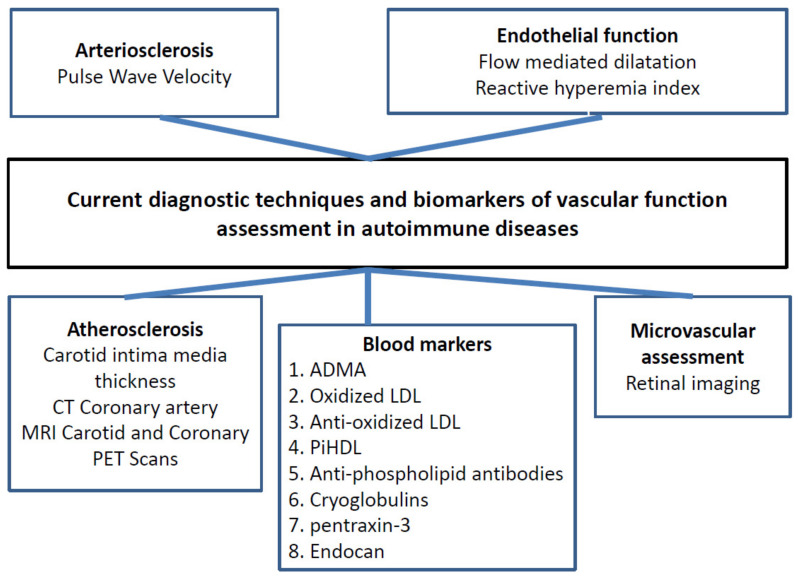
Overview of diagnostic techniques and biomarkers that are currently used in autoimmune rheumatic disease. Legend: PiHDL: proinflammatory, high-density lipoprotein; LDL: low-density lipoprotein; ADMA: asymmetric dimethylarginine; PET scan: positron emission tomography scan; FMD: flow mediated dilation; CT Scan: computerized tomography; MRI: magnetic resonance imaging.

**Table 1 jcm-10-04427-t001:** An outline of different studies that examined effects of autoimmune rheumatic diseases on vascular function and assessed risk of accompanying atherosclerosis and development of cardiovascular diseases.

Author(s)	Study Type	Methodology	Results	Interpretation
Kerola, et al., (2012) [93]	Review	Effect of anti-inflammatory drugs on FMD as well as atherosclerosis changes in carotid artery in RA	Improvement of FMD following anti-inflammatory usage for 6–12 month Carotid intima-media thickness and plaque present in first year of diagnosis	Strict control of nontraditional risk factors required to decrease cardiovascular mortality
Crowson et al., (2018) [94]	Review	Follow-up of studies for detection of cardiovascular diseases (included 5638 patients without obvious cardiovascular diseases)	30% of cardiovascular diseases developed are related to RA	Smoking and hypertension are the most important traditional risk factors related to cardiovascular diseases
Ruscitti et al., (2017) [99]	Prospective study of 347 patients with RA over one year	Follow-up for detection of cardiovascular diseases via assessment of carotid intima-media thickness and atheromatous plaques	Increased incidence of subclinical atherosclerosis at one year of follow up	Combination of traditional and nontraditional risk factors responsible for premature atherosclerosis
Maradit-Kremers et al., (2005) [104]	603 patients with RA and 603 controls, followed up until death	Follow up in out-patients clinic and hospitalized patients, to assess development of angina, myocardial infarction and sudden cardiac death	Rheumatoid arthritis patients have two times more incidence of unrecognized myocardial infarction or sudden cardiac death but not angina as compared to healthy persons	Traditional risk factors alone cannot explain the increased incidence of cardiovascular disease in patients with rheumatoid arthritis
Fan, et al., (2012) [108]	102 patients with RA and 46 healthy controls	Brachial artery FMD and carotid intima media thickness assessment	FMD was lower in rheumatoid group as compared to that of controls. Higher CIMT values seen in RA patients	No significant correlation between FMD and carotid wall thickness
Adawi et al., (2018) [109]	44 RA patients compared with healthy control	Assessment of FMD in the brachial artery	86% of the RA patients showed varying degrees of endothelial dysfunction	Early recognition—and monitoring—of endothelial dysfunction and subclinical atherosclerosis are important in follow-ups in RA patients
Kiss et al., (2006) [125]	61 SLE patients and 26 healthy controls	FMD and CIMT measurements carried out	Significant differences between SLE patients and healthy controls: Lower FMD and higher CIMT in SLE	SLE patients should be regularly screened for the development of premature atherosclerosis
Mak, et al., (2017) [126]	Case-control study with 71 SLE patients and 71 healthy controls as well as a meta-analysis of 25 case controls studies with 1313 patients and 1012 healthy controls	Endothelial dependent FMD was assessed	Lower FMD was seen in SLE patients as compared to healthy controls	Diabetes, hypertension, and renal disease have greater effect on the atherosclerotic process in SLE patients
Sacre et al., (2014) [127]	Cross-sectional controlled study in 41 SLE patients and 35 healthy controls	Carotid-femoral PWV was measured for assessment of arterial stiffness	Increased PWV in SLE patients as compared to healthy persons	Associated hypertension and corticosteroid treatments have an even greater effect on arterial stiffness in SLE patients
Thompson et al., (2008) [130]	Prospective study of 217 SLE female patients	Carotid ultrasound at baseline and follow- up for assessment of carotid plaques and CIMT	Accelerated plaque formation in SLE	Traditional and nontraditional risk factors play important roles in the progression of atherosclerosis in SLE
Yong, et al., (2019) [133]	Systematic review and meta-analysis of 8 studies in 767 PSS patients to assess arterial stiffness and subclinical atherosclerosis	PWV and intima-media thickness assessments carried out in PSS patients	PSS patients have higher PWV and intima-media thickness as compared to healthy controls	PSS patients have premature atherosclerosis. More longitudinal studies are needed to assess the time course of the development of atherosclerosis and risk of cardiovascular diseases
Sezis Demirci, et al., (2016) [134]	Arterial stiffness in PSS was assessed in 75 patients and compared with 68 healthy controls	Carotid-femoral PWV measurements were carried out	PWV was higher PSS patients as compared to healthy controls	Arterial stiffness in PSS patients may be due to the associated hypertension, steroidal usage, and hyperlipidemia in these patients, and may not be due to the disease itself

Legend: PSS: primary Sjögren’s syndrome; PWV: pulse wave velocity; CIMT: carotid intima-media thickness; FMD: flow mediated dilation; RA: rheumatoid arthritis; SLE: systemic lupus erythematosus.5. Endothelial Dysfunction and Arterial Stiffness.

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
