# Peer review of "Autoimmune Rheumatic Diseases and Vascular Function: The Concept of Autoimmune Atherosclerosis"

_jcm, 2021, doi:10.3390/jcm10194427_

Round 1
Reviewer 1 Report
Interesting paper, well organized, up-to-date.
Some minor concern about the style of references. I suggest to uniform to New England Journal of Medicine or Vancouver. To specify that the paper is in English (eng) in unnecessary.
Ref #160 includes a double name of the journal: SAMJ and S Afr Med J
More importantly, the last 7 references have a double number: (170, 178), (171, 179) and so on.
The authors should pay more attention to these formal aspects of the paper.
Reviewer 2 Report
In the current review, the authors assess a modern topic regarding the concept of Autoimmune Atherosclerosis. The paper is overall well written, but it is very narrative and the authors offer many general data but lack some consistency for proposing more mechanisms that are related to the topic. Remarks and suggestions:
- The English should be revised. Just an example:“ discussed here in detail are also the possible effects “ and many other.
- The abstract should be modified as it is too general and refers little to the topic.
- The authors should propose more details regarding the subject and check deeper for more mechanisms. In each section, there are too many general data and few paragraphs that address the topic.
- One or two figures regarding potential mechanisms would provide a better understanding for readers than just plain text.
- The authors may summarise data from studies into tables, on “Autoimmune Rheumatic Diseases and Vascular Function assessment and results” categories. The visual impact would be greater.
- The figure 1 is not displayed.
- How did the authors remove the articles for which there were only abstracts displayed if they were published in extenso? This can represent a bias.
- The abbreviations should be predefined at the first use.
- At the beginning of section 5, it is not clear what the phrases in the centre represent.
- Regarding PWV determination, there are several other devices and methods that may be used.
- The average thickness on intima-media is 0.9 mm, not 0.9 ml.
Round 2
Reviewer 2 Report
The authors have answered properly to the comments. I find the article suitable for publication.